Short-term microplastic effects on marine meiofauna abundance, diversity and community composition

de França Flávia J.L. flaviah.lobato@hotmail.com 1
Moens Tom 2
da Silva Renan B. 1
Pessoa Giovanna L. 1
França Débora A.A. 1
Dos Santos Giovanni A.P. giopaiva@hotmail.com 1
1 Campus Recife, Center for Biosciences, Department of Zoology, Universidade Federal de Pernambuco , Recife , Pernambuco , Brazil
2 Marine Biology Lab, Biology Department, Ghent University , Ghent , Flanders , Belgium
Idris Izwandy
Electronic publication date: 2024 Jul 31
Publication date: 2024
Volume: 12
Electronic Location ID: e17641
Received 2024 Jan 19; Accepted 2024 Jun 6
Copyright: ©2024 de França et al.
Copyright year: 2024
Copyright holder: de França et al.
License: This is an open access article distributed under the terms of the Creative Commons Attribution License, which permits unrestricted use, distribution, reproduction and adaptation in any medium and for any purpose provided that it is properly attributed. For attribution, the original author(s), title, publication source (PeerJ) and either DOI or URL of the article must be cited.
License URL: https://creativecommons.org/licenses/by/4.0/

Keywords: Meiobenthos, Pollution, Marine impact, Dose-dependent effect, Plastic, Microcosm, Fluorescence, Ingestion, Community structure

Funding: The Coordenação de Aperfeiçoamento de Pessoal de Nível Superior—Brasil (CAPES)—Finance Code 001 The Fundação Coordenação de Aperfeiçoamento de Pessoal de Nível Superior—CAPES 88887.635950/2021-00 The Fundação de Amparo à Ciência e Tecnologia do Estado de Pernambuco—FACEPE IBPG-0102-2.04/23 BIC from PROPESQI 230121628 CAPES 88887.702887/2022-00 PROPESQI Notiz number 09/2019 from the Federal University of Pernambuco HOTMIC (BEL.BRA.2019.0008.01) in the framework of the JPI Oceans Joint Action on Ecological Aspects of Microplastics This study was financed by the Coordenação de Aperfeiçoamento de Pessoal de Nível Superior—Brasil (CAPES)—Finance Code 001. Flávia J L de França was supported by grant number 88887.635950/2021-00 from the Fundação Coordenação de Aperfeiçoamento de Pessoal de Nível Superior—CAPES. Renan B da Silva was supported by grant number IBPG-0102-2.04/23 from the Fundação de Amparo à Ciência e Tecnologia do Estado de Pernambuco –FACEPE. Giovanna L. Pessoa was supported by BIC grant number 230121628 from PROPESQI. Débora AA França was supported by grant number 88887.702887/2022-00 from CAPES. Giovanni AP Dos Santos was supported by PROPESQI Notiz number 09/2019 from the Federal University of Pernambuco. Additional funding was obtained from the project HOTMIC (BEL.BRA.2019.0008.01) in the framework of the JPI Oceans Joint Action on Ecological Aspects of Microplastics. There was no additional external funding received for this study. The funders had no role in study design, data collection and analysis, decision to publish, or preparation of the manuscript.

==============================
Background

Due to the copious disposal of plastics, marine ecosystems receive a large part of this waste. Microplastics (MPs) are solid particles smaller than 5 millimeters in size. Among the plastic polymers, polystyrene (PS) is one of the most commonly used and discarded. Due to its density being greater than that of water, it accumulates in marine sediments, potentially affecting benthic communities. This study investigated the ingestion of MP and their effect on the meiofauna community of a sandy beach. Meiofauna are an important trophic link between the basal and higher trophic levels of sedimentary food webs and may therefore be substantially involved in trophic transfer of MP and their associated compounds.

Methods

We incubated microcosms without addition of MP (controls) and treatments contaminated with PS MP (1-µm) in marine sediments at three nominal concentrations (103, 105, 107particles/mL), for nine days, and sampled for meiofauna with collections every three days. At each sampling time, meiofauna were collected, quantified and identified to higher-taxon level, and ingestion of MP was quantified under an epifluorescence microscope.

Results

Except for Tardigrada, all meiofauna taxa (Nematoda, turbellarians, Copepoda, Nauplii, Acari and Gastrotricha) ingested MP. Absorption was strongly dose dependent, being highest at 107 particles/mL, very low at 105 particles/mL and non-demonstrable at 103 particles/mL. Nematodes accumulated MP mainly in the intestine; MP abundance in the intestine increased with increasing incubation time. The total meiofauna density and species richness were significantly lower at the lowest MP concentration, while at the highest concentration these parameters were very similar to the control. In contrast, Shannon-Wiener diversity and evenness were greater in treatments with low MP concentration. However, these results should be interpreted with caution because of the low meiofauna abundances at the lower two MP concentrations.

Conclusion

At the highest MP concentration, abundance, taxonomic diversity and community structure of a beach meiofauna community were not significantly affected, suggesting that MP effects on meiofauna are at most subtle. However, lower MP concentrations did cause substantial declines in abundance and diversity, in line with previous studies at the population and community level. While we can only speculate on the underlying mechanism(s) of this counterintuitive response, results suggest that further research is needed to better understand MP effects on marine benthic communities.

Introduction

Global plastic production has surpassed 400 million tonnes annually since 2022 (Plastics-the Facts, 2022). Due to the inefficiency of policies in promoting proper waste disposal, a substantial amount of plastic ends up in marine environments (Jambeck et al., 2015; Wayman & Niemann, 2021) where it undergoes degradation, giving rise to microplastics (MPs) (Browne et al., 2011; Thompson et al., 2009). These are solid polymer-based particles measuring less than 5 mm in length (Arthur, Baker & Bamford, 2009). Many plastic polymers have a specific density higher than water (>1 g/cm3) (Van Cauwenberghe et al., 2015). This property prompts their preferential deposition, primarily in sediments (Hoseini & Bond, 2022). Biofouling further increases the density of these particles (Galloway, Cole & Lewis, 2017; Harrison et al., 2018; Kaiser, Kowalski & Waniek, 2017). Consequently, microplastic abundances tend to be higher in sediments than in the water column (Harris, 2020; Hoseini & Bond, 2022; Scherer et al., 2020) and benthic organisms are exposed to higher microplastic loads compared to plankton (Haegerbaeumer et al., 2019).

Polystyrene (PS) is one of the most extensively utilized plastic polymers globally and has a density that is slightly greater than that of water (1.05 g/cm3) (Plastics-the Facts, 2022). Aside from inducing mechanical effects through ingestion (Lee et al., 2013; Yu et al., 2020), PS also inflicts various damages on non-ingesting benthic invertebrates, impacting their survival rate (Bejgarn et al., 2015; Gewert, MacLeod & Breitholtz, 2021), reproduction (Mueller et al., 2020; Schöpfer et al., 2020; Yu et al., 2020), growth (Lei et al., 2018; Shang et al., 2020), and physiological well-being (Acosta-Coley et al., 2019; Lei et al., 2018). Moreover, besides the observed effects on individual groups, substantial impacts have been found at the community level, manifested as a decline in the abundance and/or taxonomic richness of benthic animals (Corinaldesi et al., 2022; Wakkaf et al., 2020).

Meiofauna, a consortium of minute metazoans passing through a sieve with apertures of one mm but being retained on a sieve with openings of 38 µm, encompasses juvenile macrofauna as well as organisms that undergo their entire life cycle dependent on the substrate (Giere, 2009). Due to their high abundances and functional and structural diversity, they may play pivotal roles in marine benthic ecosystems (Schratzberger & Ingels, 2018). They (micro) bioturbate sediments (Cullen, 1973) and form trophic links between lower trophic levels and higher ones, thereby influencing the cycling of organic matter and impacting the physical, chemical, and biological properties of the sediment (Schratzberger & Ingels, 2018). Furthermore, meiofauna exhibit a remarkable diversity of animals with distinct phylogenies, functional characteristics and environmental sensitivities (Giere, 2009; Pusceddu et al., 2007; Santos et al., 2018; Schratzberger & Ingels, 2018). In addition, due to characteristics such as a limited mobility and hence escape capacity and a short life cycle, they serve as a very useful tool for comprehending environmental impacts (Giere, 2009; Schratzberger & Ingels, 2018).

To date, studies on the effects of MP on meiofauna have predominantly focused on nematodes, the predominant taxonomic group among the meiofauna, but only few of these have investigated community structure and diversity (Allouche et al., 2022; Allouche et al., 2021; Wakkaf et al., 2020). Even though only a few studies have examined the differential effects of microplastics (MP) on various meiofauna groups, such analysis is crucial. Despite limited evidence, microplastics have already been discovered in natural meiofauna populations (Gusmao et al., 2016). There is a need for an efficient analysis to address uncertainties regarding whether meiofauna in situ can truly ingest large quantities of microplastics. Current advancements in microplastics investigation techniques have been unable to conclusively prove or detect meiofauna ingestion of very small microplastic particles (Todaro et al., 2023). Even fewer studies have delved into dose–response effects on meiofauna community structure, with a limited examination of response variables beyond total abundance (Rauchschwalbe et al., 2022a; Wakkaf et al., 2020). Adverse effects of MP at environmentally relevant abundances on aquatic invertebrates have been demonstrated, including life-history parameters such as survival (Leung & Chan, 2018), reproduction (Ogonowski et al., 2016), and growth (Zimmermann et al., 2020). Another aspect that has been only minimally explored in studies involving meiofauna and microplastics is the relationship between exposure time and impact. While some studies with meiofauna effectively establish this factor (Fueser, Mueller & Traunspurger, 2020a; Mueller et al., 2020), they have not yet done so at the community level.

The objective of this work is to investigate ingestion of microplastics by, and short-term effects of exposure to microplastics on, marine meiofauna abundance, taxonomic diversity and community structure, with emphasis on (a) the dose-dependence of the response, and (b) the importance of incubation time. In line with this objective, we hypothesize that: H1: Any effects of microplastics on meiofauna communities will reduce their abundance and taxonomic diversity; and H2: Effect sizes will become larger with increasing dose and exposure time.

Materials & Methods

Study area and sampling

Meiofauna samples were collected at low tide on April 14, 2022, at Praia de Cupe in the City of Ipojuca, Pernambuco, Brazil (8°27′29.4″S 34°59′03.2″W). This sandy beach features fine sediments and a robust presence of reefs and pools. Samples were collected non-quantitatively by scraping the upper two cm of sediment, where the highest abundances of meiofauna occur (Coull & Chandler, 1992). This sediment was then carefully homogenized using a shovel and transported to the laboratory in buckets with natural seawater. An air pump was used to maintain oxygenation during transport. In the laboratory, the samples were subjected to environmental conditions with constant temperature and salinity (28 °C and 35, respectively) for a 7-day period to ensure stabilization of the fauna before initiating the experiment (Monteiro et al., 2019; Vafeiadou et al., 2018). At the sampling location, we also collected four replicate 10-cm2 Perspex cores as ambient ‘controls’ (Camb) representing the natural sediment meiofauna community at time and place of sampling. These were preserved in a 4% buffered formaldehyde solution. After stabilization of the sediment to be used in the experimental incubations, four additional replicate cores of 10 cm2 each were taken from this sediment and preserved in formaldehyde to provide a T0 representing the meiofauna community at the start of the experimental incubation.

Sampling was carried out in public beaches, and the field study, which did not involve endangered species, required no special permissions or permits due to the microscopic, non-pathogenic nature of meiofauna. Additionally, none of the meiofauna species are subject to special conservation concerns.

Microcosm setup

The microcosm design and incubation were as described in Vafeiadou et al. (2018). Briefly, tanks with distilled water, monitored with thermostats, served as thermal regulators for 1-L beakers (dimensions: 18 cm high and 11 cm diameter). Inside each beaker, a 5–6-cm deep sediment layer was submerged in seawater and oxygenated by an aeration system that cycled air through the sediment through negative pressure, designed to prevent anoxia throughout the sediment column (Vafeiadou et al., 2018). Each microcosm received 300 g of ‘natural’ sediment containing fauna and 100 g of defaunated sediment with microplastics (see Section ‘Microplastics treatments’). These two sediment volumes were carefully and thoroughly homogenized by mixing them by hand. The microcosms were conditioned at 28 ± 1 °C and a salinity of 35 ± 1, and monitored daily for 9 days. There were four replicate microcosms per treatment.

Microplastics treatments

We used 1-µm diameter fluorescent (Fluoresbrite®; excitation maximum: 441 nm, emission maximum: 485 nm) polystyrene microspheres (Polysciences Europe GmbH, Baden-Wuerttemberg, Germany). Considering the ongoing debate surrounding the lower limit of microplastics (MP) in the literature (Arthur, Baker & Bamford, 2009; Bermúdez & Swarzenski, 2021), we want to clarify that while the size of the MP tested in this study is typically categorized as nanoplastics, we acknowledges the ambiguity surrounding their classification and avoid potential complications or biases that could arise from attempting to make a clear distinction, so we have chosen not to differentiate between nanoplastics and microplastics for the purposes of this research. The experimental design comprised four treatments: experimental control (C) without addition of MP; low concentration (103 part./mL); medium concentration (105 part./mL); high concentration (107 part./mL). Low and medium concentration both represent concentrations that are commonly found in marine benthic environments (Harris, 2020; Hoseini & Bond, 2022), whereas the high concentration is currently in excess of known field concentrations but may represent a future scenario of increasing microplastic pollution.

Microplastic concentrations were prepared by diluting a stock concentration of 4.55 × 1010 particles/mL (part./mL) with distilled water and thoroughly mixing it in the appropriate amount with defaunated sediment to create sediment with a concentration of 400 × 107 part./ 100 g sediment. Defaunation was done using three cycles of freezing (−20 °C, 12h) and thawing (18 °C, 48h). A total of 100 g of this defaunated sediment polluted with MP was then thoroughly mixed with 300 g of natural sediment, resulting in our higher sediment concentration of 107 MP part./mL. The lower two nominal MP concentrations (105 and 103 part./mL) were then obtained from diluting the polluted defaunated sediment (100 g of sediment at 107 part./mL) with MP-free defaunated sediment; therefore, a serial dilution was performed keeping the proportion of 100 g of defaunated sediment to 300 g of natural sediment for all experimental concentrations.

Laboratory sampling and sample processing

The experimental microcosms were sampled after three (T3), six (T6) and nine days (T9) of incubation. Samples were collected using transparent Perspex corers with an inner diameter of 3.6 cm. All treatments were replicated four times, but at T9 a replicate of 105 part./mL was lost. The samples were rinsed with a jet of tap water over a 300-µm and a 38-µm mesh sieve, and meiofauna was extracted using the centrifugation-flotation method (Murrell & Fleeger, 1989). Subsequently, the meiofauna was frozen (−20 °C, 12h) while still alive to avoid egestion and defaecation (Moens, Verbeeck & Vincx, 1999) and then preserved in 4% formaldehyde; they were stained with rose bengal to facilitate counting, and identified to phylum or ‘major taxon’ level using a stereomicroscope (Zeiss, Oberkochen, Germany, Stemi 305). Approximately 130 nematodes and 30 other meiofauna were haphazardly picked up from the samples and mounted on slides for epifluorescence microscopy to record ingestion of MP. The mounting process comprised three stages: (I) After their removal from the sample, specimens were placed (12h) in an embryodish containing a solution composed of 99% formaldehyde and 1% glycerin in a desiccator. (II) Subsequently, four drops of a second solution composed of 95% alcohol and 5% glycerin were added in the same embryo dish, and this was repeated 4 times at 2-h intervals. (III) In a last step, a solution of 50% ethanol and 50% glycerin was added (Grisse, 1969). MP inside the gut or stuck to the body surface were counted using an epifluorescence equipped optical microscope (Zeiss, Oberkochen, Germany, Scope.A1). For nematodes, MP were separately quantified for three body regions: anterior (oral cavity to the end of the pharynx), median (end of the pharynx to the (posterior) ovary or testis), and posterior (end of the (posterior) ovary or testis to the anus).

Data analysis

The effect of different MP concentrations on the taxonomic composition of meiofauna communities was analysed using a non-metric multidimensional scaling (nMDS) analysis on fourth-root transformed density data (to reduce differences between abundant and rare taxa (Clarke & Gorley, 2015), the similarity matrix was calculated using the Bray-Curtis index. In order to identify the taxa responsible for the (dis)similarity between and within treatments, a similarity percentage analysis (SIMPER) was performed. Effects of microplastics and incubation time on meiofauna major taxon abundance and diversity were analysed using a univariate analysis of variance. For this, the density data were fourth-root transformed and the similarity matrix was constructed using Euclidean Distance. The diversity indices used were richness, evenness (Pielou’s J) and diversity (Shannon-Wiener H’). We performed a two-way Multivariate Permutational Analysis of Variance (Permanova) to establish the significance of the detected trends observed in the nMDS and SIMPER analyses, and to understand the significant differences between the abundance and diversity index (Anderson, 2008). Furthermore, Permutational Analysis of Multivariate Dispersion (PERMDISP) was applied to test the homogeneity of the data. To examine particle ingestion in each body region of the nematode, considering time and treatment, the raw data on ingested particles were utilized to generate a shade graph. In this shade graph, each spot corresponds to the quantity of ingested particles. Univariate and multivariate analyses, shade graphs, and non-Metric Multidimensional Scaling (nMDS) were conducted using PRIMER 7.1 software with PERMANOVA add-on. Total meiofauna and nematode density data, diversity indices, and meiofauna ingestion data were graphically represented using Sigmaplot 12.5 software.

Results

Effect of short exposure to different concentrations of microplastics on the meiofauna community

Total meiofauna density

Although meiofauna abundances at T0 were significantly lower than in the field control (Camb) (Pseudo-F = 43.134; p = 0.0009), taxonomic richness did not differ significantly between the two (Pseudo-F = 4.6156; p = 0.0756). Moreover, during the 9-day incubation, the abundance of meiofauna did not differ between T0 and experimental controls (Pseudo-F = 1.6155; p = 0.2403) (Fig. 1A).

Figure 1 Total meiofauna density (A) and nematode density (B) (both in ind/10 cm2).

The red line represents the average of four replicates, and the black line represents the median. Field control (Camb), control after stabilization (T0), and treatments (C: no MP, and three treatments with different MP concentrations) after 3 (T3), 6 (T6), and 9 (T9) days of incubation. Capital letters represent significant differences between times for each treatment, while small letters represent significant differences between treatments within each experimental time. Concentrations in particles/mL.

A total of 5,629 individuals were identified to major taxon level in the experiment, distributed over ten meiofaunal taxa: Nematoda, Oligochaeta, Ostracoda, Gastrotricha, Copepoda, turbellarians, Tardigrada, Nauplii, Acari and Polychaeta. Nematoda was by far the most abundant taxon with 4,564 individuals, followed by turbellarians (N = 595).

The total density of meiofauna (Fig. 1A) differed significantly across time (Pseudo-F =22.91; p = 0.0001; PERMDISP: p = 0.0187), treatment (Pseudo-F = 41.57; p = 0 .0001; PERMDISP: p = 0.0537), and time vs treatment. (Pseudo =F = 7.67; p < 0.0003). Surprisingly, the treatment with the low particle concentration (103 part./mL) had a significantly lower meiofauna density than the high concentration treatment (p < 0.0004). Differences between treatments were most pronounced after 3 days of incubation, when all MP treatments differed significantly from each other and from the control (Fig. 1A). From 6 days onwards, the high MP treatment no longer differed from the control, whereas the medium MP treatment alternately resembled the control or the low MP treatment (Fig. 1A). The latter consistently exhibited the lowest meiofauna abundances, which were always significantly lower than the high MP treatment and the control (Fig. 1A). With the exception of the field control, all samples (T0 and treatments) were consistently dominated by Nematoda (Fig. 1B), hence the same abundance patterns and trends appeared in total meiofauna and nematodes.

Over time, experimental controls did not show any difference in meiofauna density (p < 0.2852). In the treatment with the lowest concentration of MP (103 part./mL), despite having lower densities compared to the other treatments, the density of meiofauna increased with time (p < 0.0348). In the treatment with the medium concentration of particles (105 part./mL), as in the controls, no significant variation in meiofauna density over time was observed (p < 0.9168) (Fig. 1A). The same pattern observed for the total density of meiofauna was observed for the density of nematodes, given the predominance of this taxon in all treatments (Fig. 1B).

Community structure and diversity

Meiofauna richness at higher taxon level differed between treatments (Pseudo-F =8.43; p = 0.0024; PERMDISP: p = 0.026) and time vs treatment (Pseudo-F = 9.47; p = 0.0001). Time alone displayed a borderline significant effect (Pseudo-F = 3.3849; p = 0.0531; PERMDISP: p = 0.0003). At T3, the low and medium MP concentrations had a lower richness than the control due to the disappearance of Ostracoda, Acari (both p < 0.0059) and Tardigrada at the lowest MP concentration. At T6, the high MP concentration exhibited a higher richness than the control (p = 0.0063) due to the presence of Ostracoda, Oligochaeta and Polychaeta. Finally, at T9, richness did not differ significantly between treatments. This was not due to a decrease in richness in one or more treatments, but because of a higher richness than at earlier sampling days in the low and medium treatments (Fig. 2). This indicates that the lower richness after 3 and 6 days in these two treatments was likely a consequence of undersampling due to the low meiofauna abundances.

Figure 2 Meiofauna relative abundance and richness.

A multicolored bar represents the relative abundance (%) of meiofauna taxa on the left Y-axis, while points with whisker bars represent taxon richness on the right Y-axis. Field control (Camb), control after stabilization (T0), collection after 3 (T3), 6 (T6), and 9 (T9) days of exposure to MP. Concentrations in particles/mL.

Evenness of meiofauna taxa differed between times (Pseudo-F = 12.709; p = 0.0003; PERMDISP: p = 0.5215) and treatments (Pseudo-F = 23.048; p = 0.0001; PERMDISP: p = 0.9119), but not as a function of the interaction of time vs treatment (Pseudo-F = 0.77457; p = 0.5953). The highest evenness occurred after 3 days (p < 0.004) (Fig. S1), after which it declined as a result of the strong dominance of a single taxon (Nematoda). Across times, but most pronounced at T3, the low MP treatment had a more even meiofauna community than the control and other MP concentrations (all p < 0.0083). The high evenness in the treatment with the lowest concentration of MP (103 part./mL) may be linked to the low abundances of meiofauna taxa that dominated the other treatments.

Only the factor treatment significantly affected Shannon diversity (H’) (Fig. S2) (Pseudo-F = 5.9956; p = 0.012; PERMDISP: p = 0.3243), being higher at the lower two MP concentrations (p < 0.0364). In the pairwise comparisons at both T6 and T9, the treatment with the lowest particle concentration (103 part./mL) had a higher H’ compared to the control and the other treatments (p < 0.0374).

The structure of the meiofauna community (Fig. 3) differed significantly across time (Pseudo-F =4.5979; p = 0.0001; PERMDISP: p = 0.0018), treatment (Pseudo-F =4.0359; p = 0.0003; PERMDISP: p = 0.0004), and time vs treatment (Pseudo-F =2.1867; p = 0.0062). In the treatment with a concentration of 107 part./mL, meiofauna community structure was not significantly different from that in the control throughout the experiment (p = 0.6975). By contrast, the low MP treatment had a significantly different community structure than the control throughout the experiment (p < 0.0499) and the high treatment (p < 0.0283) up to six days of incubation. The response of the 105 part./mL MP treatment demonstrated an intermediate pattern. It closely resembled the low MP treatment after 3 days, but progressively aligned more with the control and high MP treatment from T6 onwards. Overall, as time progressed, the dissimilarity between the treatments diminished.

Figure 3 Non-metric multidimensional scaling (nMDS) of meiofauna community composition in four different treatments (C, 103, 105, 107 part./mL) at three incubation times (T3, T6 and T9).

The nMDS is based on meiofauna taxon densities (fourth-root transformed) and using Bray-Curtis similarity. Vectors shown are correlation vectors for the different meiofauna taxa, where the size of the vector indicates the value of the Pearson correlation.

Despite a greater similarity of the fauna between 105, 107 part./mL, and C, it was still noticeable that a gradient was formed, in which 107 part./mL and C were most similar, followed by 105 part./mL, and finally, 103 part./mL, which presents a structure of the meiofauna community that is less similar to all others (Fig. 3).

SIMPER analysis demonstrated that the control treatment had the highest similarity among replicates (79.2%), followed closely by 107 part./mL (76.0%) and 105 part./mL (75.7%), but differing distinctly from the much more variable 103 part./mL (53.9%). Nematoda was the taxon that most contributed to the similarity of the fauna, with its contribution decreasing with increasing microplastic concentration. The second taxon that contributed most to the similarity were turbellarians at all MP concentrations, except for 103 part./mL, where the second highest contributor was Gastrotricha with 26.6%.

The greatest dissimilarities in taxon composition between treatments were seen between 103 and 107 part./mL and between treatments C and 103 part./mL (43.2 and 42.0% dissimilarity, respectively). In both comparisons of treatments, Nematoda and turbellarians were the main contributors to the dissimilarity. The dissimilarities between C and the medium and high concentrations of MP (105 and 107 part./mL) did not exceed 26%. Nematoda and Copepoda were the taxa that most contributed to the dissimilarity between C and 105 part./mL (39.7%), while between C and 107 part./mL, Nauplii and Copepoda contributed the most (33.8%).

Ingestion of particles by meiofauna

Seven meiofauna groups, including the phyla Nematoda, Tardigrada and Gastrotricha, turbellarians, Copepoda, Acari and crustacean larvae (Nauplii), were assessed for both internal (ingestion) and external microplastic ‘uptake’ (Fig. 4) (Figs. S3A–S3F and S4). At the low MP concentration (103 part./mL), no ingestion occurred, and microplastic particles were also not detected externally. At the medium MP concentration (10 5 part./mL), only Nematoda and turbellarians exhibited microplastic ingestion, albeit at very low levels (on average 1.4 and 1.0 particles per individual, respectively). Copepoda at this MP concentration did not exhibit ingestion but carried an average of three external particles per individual (SE ± 1.0). The highest ingestion occurred at the high MP concentration (107 part./mL), where all meiofauna groups ingested microplastics, except Tardigrada, which only exhibited external contamination (on average 14.1 part./ind. (SE ± 5.33). Acari (16.4 ± 5.47 particles per individual) and Copepoda (15.6 ± 6.53 particles per individual) exhibited the highest particle ingestion (Figs. S3B–S3F), followed by Nematoda (9.9 ± 1.24 particles per individual). It is important to note that the sample size for Acari and Copepoda was low.

Figure 4 Meiofauna Internal (ingestion) and external contamination by microplastic particles at various concentrations.

The left Y-axis represents the percentage of internal and external particles per individual, while the right Y-axis illustrates the average (±SE) count of internal and external particles per individual for each evaluated meiofauna group at each tested concentration. MP concentrations in particles/mL.

The internal and external contamination of meiofauna groups did not differ between exposure times (Pseudo-F = 1.1765; p = 0.3841) but displayed a significant variability caused by treatment (Pseudo-F = 5.0052; p = 0.0206). At low MP concentration, neither internal nor external contamination was observed in any meiofauna group. At the medium concentration, 0.98% of Nematoda and 2.5% of turbellarians ingested microplastics. 2.5% of turbellarians and 25% of copepods were externally contaminated (Figs. S3A–S3D). At high MP concentration, a significantly higher internal and external contamination was observed compared to other treatments (p < 0.0399). Over 50% of nematodes ingested microplastics, followed by Nauplii (44.4%) and Gastrotricha (38.7%). Copepoda and turbellarians had more individuals externally contaminated (36.5% and 27.8%, respectively) than internally contaminated (31.7% and 20.2%, respectively). Acari had an equal number of internally and externally contaminated individuals (35.7%). Tardigrada did not exhibit microplastic ingestion but 54% of individuals showed external particle presence, mainly in the locomotory appendages (Fig. S3A).

At medium MP concentration, particles were observed solely in the medial body region, specifically within the intestine. Conversely, at the highest concentration, MP were observed from the oral cavity to the end of the intestine (Fig. S4), with an increase as experimental duration progressed (Fig. 5).

Discussion

Microplastic ingestion by different meiofaunal taxa

All meiofauna groups examined here except Tardigrada ingested microplastics in our experimental setting. To the best of our knowledge, this study thus marks the first documentation of microplastic ingestion in turbellarians and Gastrotricha. Turbellarians can prey on nematodes, copepods, and even individuals of the same taxonomic group (Giere, 2009). It is therefore plausible that at least part of the ingested MP derived from their prey, indicating that microplastics can indeed be transferred up the trophic chain (Costa et al., 2020). On the other hand, Gastrotricha are microphagous, primarily feeding on bacteria and protozoans (Giere, 2009; Todaro & Luporini, 2022). It is plausible that Gastrotricha accidentally co-ingested microplastics of similar size and shape as their natural food. In addition, the presence of microorganisms growing on the surface of microplastics (Amelia et al., 2021; Haegerbaeumer et al., 2019) may have contributed to this MP ingestion (Fabra et al., 2021; Murano et al., 2021), particularly sincean incubation of 9 days is more than enough time for bacterial growth to occur on the surface of MP (De Tender et al., 2017; Tu et al., 2020). The absence of MP ingestion by Tardigrada likely links to the structure of their feeding apparatus, which includes a mouth tube with a stylet used to pierce and suck rather than ingest prey organisms whole (Guidetti et al., 2012). However, 54% of tardigrades in this study had particles adhering to their bodies.

Figure 5 Total amount of internal particles in Nematoda by body region (pharyngeal, middle gut and posterior intestine), treatment and exposure time.

The color of the 0-1,000 scale represents the amount of particles ingested. Each color represents concentration in part./mL.

Copepods, along with nematodes, are the meiofauna group that have been most extensively examined for uptake and effects of microplastics (Li et al., 2020; Sun et al., 2021; Yang et al., 2022). Despite some monospecific studies with benthic copepods reporting substantial MP ingestion (Quanbin et al., 2020; Todaro et al., 2023; Xie et al., 2022), and even a preference for MP over food particles (Lee et al., 2013), ingestion in this study was limited, occurring only at the highest MP concentration, partly in line with (Di Lorenzo et al., 2023; Fueser, Mueller & Traunspurger, 2020a; Fueser, Mueller & Traunspurger, 2020b) who reported copepods as the meiobenthic taxon with the lowest ingestion. At the same time, however, approximately 25% of copepods in our study had particles affixed to their locomotory appendages, a phenomenon also observed in pelagic copepods (Quanbin et al., 2020). The presence of MP in the appendages may disturb their locomotion, for instance through a reduction in swimming speed (Suwaki, De-La-Cruz & Lopes, 2020).

Nematodes exhibited the second- or third-highest MP ingestion when exposed to 105 and 107 part./mL, respectively. Free-living nematodes can ingest particles large enough to trigger a feeding response (Moens & Vincx, 1997). Many bacterivores, for instance, can ingest particles larger than 0.5 µm but with an upper limit of 6 µm (Fueser et al., 2019), whereas nematodes with other feeding strategies may also ingest considerably larger-sized particles (Moens et al., 2014). These particles accumulate primarily in the pharynx and intestine of nematodes (Fueser et al., 2022; Fueser et al., 2019; Huang et al., 2023; this study). Numbers of particles in the worms’ guts tend to rise with exposure time (Quanbin et al., 2020; Shang et al., 2020; Zhang et al., 2019; this study), albeit to a limited extent. Note that our ingestion results are conservative because elutriation and preservation of live meiofauna from sediments may trigger egestion and/or defaecation, even though we adopted a protocol designed to limit such effects (Moens, Verbeeck & Vincx, 1999).

MP ingestion was most pronounced at the highest MP concentration, as expected, minimal at medium and non-existent at low concentration. With an average individual ingestion of <20 particles at the high MP concentration, and under the assumption that MP are not preferentially selected and ingested, one could theoretically expect <0.2 part./ind. at the medium particle concentration and even 100 times less at the low MP concentration. This pretty much aligns with our results, so ingestion followed a logical particle density-dependent pattern (Fueser, Mueller & Traunspurger, 2020a; Fueser et al., 2019; Xie et al., 2022).

Microplastic effects on meiofauna abundance

The impact of particle concentration on meiofauna density exhibited a much less expected pattern than it did on ingestion: it was more pronounced at the lowest MP concentration, intermediate at medium concentration and (nearly) absent at high concentration. These observations align with some previous findings on meiofauna biomass and abundance, where the effects were more pronounced at lower concentrations (Wakkaf et al., 2020), but contrast with others (see further). These results, together with the data on ingestion, suggest that what most impacted the meiofauna was not an immediate ‘pollution’ by microplastics. Otherwise, at the highest MP concentration, there would also have been significant reductions in meiofauna abundance. Other previous studies reported decreased meiofauna densities in the presence of higher microplastic concentrations (Allouche et al., 2022; Allouche et al., 2021; Hedfi et al., 2022). It is important to note, though, that the plastic polymers and sizes used in previous studies were different from the ones investigated here (PVC, <40 µm). While our experimental design does not allow to pinpoint the underlying mechanism, we suggest that microplastics may have induced an avoidance response, resulting in niche contraction and subsequent metabolic exhaustion (Buck, Weinstein & Young, 2018; Weihs & Webb, 1984). At 107 part./mL, meiofauna had no means of avoiding MP due to their high concentrations. Consequently, the animals inevitably ingested MP as well as organic particles and microorganisms adhering to their surface, i.e., the so-called plastisphere (Amelia et al., 2021; Haegerbaeumer et al., 2019). These MP-associated food particles may have enhanced food absorption (Gago et al., 2020; Lu et al., 2016; Qi et al., 2020; Shen et al., 2019) and may in this way have counteracted any short-term negative effects of MP on meiofauna. Notably, in Nematoda, exposure of microplastics has been observed to elevate pharyngeal pumping rate (Fueser et al., 2021), and this effect is more pronounced upon exposure to 1-µm PS particles than when these PS particles are replaced by additional bacterial cells (Fueser, Mueller & Traunspurger, 2020b; Rauchschwalbe et al., 2021). Admittedly, this is not the most parsimonious explanation for the lack of a difference in meiofauna abundances between controls and the high-particle treatment, the simplest explanation being that MP simply have no negative effects on meiofauna, but this is at odds with our results at lower MP concentrations.

Microplastic effects on taxonomic diversity and community composition of meiofauna

Anthropogenic pollution typically causes a decrease in taxonomic diversity of faunal communities, consistently so for different aspects of diversity such as richness and evenness (Corinaldesi et al., 2022; da Silva et al., 2022; Monteiro et al., 2019). In the present study, however, richness on the one hand, and evenness and Shannon-Wiener diversity on the other, behaved oppositely. Taxonomic richness behaved in a similar way as abundance, with lower values at low and medium than at high MP concentrations. Interestingly, however, richness in these treatments recovered to the level of controls by the end of the experiment. This strongly suggests that we underestimated richness in these treatments at the first sampling. This could be a result of the very low abundances of meiofauna at that time, leading to different combinations of taxa being represented in different replicates, which in turn may relate to the explanations discussed above. Evenness and Shannon-Wiener diversity, by contrast, were highest at the low MP concentration. Caution is due when interpreting these results, because the low and medium concentration treatments were characterized by very low abundances of organisms, which may well have hampered meaningful calculations of diversity and evenness. On the other hand, one might then have expected large variances around the mean, which was not the case. In any case, the different diversity indices did not exhibit typical patterns observed for communities exposed to anthropogenic impacts.

The drop in taxonomic richness at the low and medium MP concentration at T3 was partly explained by the consistent absence of Ostracoda and Acari in all replicates. While a similar drop in taxonomic richness was also observed by Corinaldesi et al. (2022) after a 3-day incubation with MP, Ostracoda and Acari then flourished, while other taxa such as Tardigrada, Polychaeta and Oligochaeta vanished. Ostracoda are often considered sensitive to anthropogenic impacts (Pusceddu et al., 2007), which might account for their disappearance at low MP concentration in our experiment. Tardigrada, despite sometimes being considered as sensitive (Corinaldesi et al., 2022; Pusceddu et al., 2007), persisted in the highest concentrations in this study.

The counterintuitive response of a pronounced community impact (decreased abundance and taxonomic richness) at low rather than high anthropogenic pressure may be attributed to a behavioral response (Avoidance theory—see Section ‘Microplastic effects on meiofauna abundance’), or to an ecotoxicological response. Our results could potentially indicate a hormetic response, where organisms exposed to low or intermediate levels of pollution ‘prepare’ for possible higher stressor levels. While the concept of hormesis is generally considered at the individual or population level (Calabrese & Baldwin, 2002), it has also been applied at the community or ecosystem level (Erofeeva, 2022). A possible negative hormetic response has already been observed at the population level in nematodes exposed to low/intermediate MP concentrations (Shang et al., 2020), whereas the present study adds to Wakkaf et al. (2020) in showing similar responses at the community level. This may be related to the roles that each meiofauna group plays within the community and/ or to their differential sensitivity to MPs (Erofeeva, 2022).

Methodological considerations

Our endeavor was to conduct this study in as realistically a manner as possible to comprehend not only the effects of microplastics on meiofaunal abundances, but also their influence on the meiofaunal community structure and diversity. The lower two concentrations used in this study bear profound environmental relevance, given their prevalence in aquatic ecosystems (Harris, 2020; Hoseini & Bond, 2022; Pabortsava & Lampitt, 2020; Scherer et al., 2020). While there are some studies on meiofauna investigating these or similar MP concentrations, most studies predominantly focus on particle ingestion (Fueser, Mueller & Traunspurger, 2020a) and/or on freshwater meiofauna (Rauchschwalbe et al., 2022b). To date, we know of only two studies which have concentrated on MP effects on marine meiofauna. However, one study examined only a single MP concentration after a single incubation time (72 h), during which substantial richness effects were observed in the control treatments (Corinaldesi et al., 2022), thus permitting only limited conclusions. The other study primarily focused on MP ingestion in relation to the interaction between bacteria and microplastics (Ridall, Asgari & Ingels, 2023). Therefore, our study represents the first exploration of marine meiofauna that centers on the analysis of community-wide responses. Additionally, it delves into the examination of ingestion patterns across different groups of marine meiofauna at different MP concentrations.

The abundance, taxonomic diversity and community composition in the environmental control (Camb) differed considerably from those in the control after sediment stabilization in the laboratory (T0), consistent with prior microcosm experiments with meiofauna (Gingold, Moens & Rocha-Olivares, 2013; Vafeiadou et al., 2018). While this calls for some caution when extrapolating results from laboratory experiments to the field, maintaining a stable community structure in control treatments throughout the experimental incubation is more crucial. Here, no significant loss in abundance and taxonomic richness was observed in the controls, indicating that the experimental incubation per se had no pronounced impact on the fauna.

Conclusions

Although the agenda that addresses (micro)plastics and their impact on the marine environment has gained prominence in recent years, there remains a lack of impact studies targeting organisms near the basis of marine benthic food webs. In this study, we observed that environmentally relevant concentrations (103 part./mL and 105 part./mL) resulted in a decrease in total meiofauna density and species richness. On the other hand, a much higher MP concentration (107 part./mL) did not differ from a control without MP addition. The maintenance of the faunal community (density and richness) at the high MP concentration may have been influenced by the feeding stimulus induced by microplastics. At the low MP concentration, the lower densities and richness compared to the control may have occurred due to a meiofauna strategy of avoiding microplastics. This strategy may have generated metabolic wear and tear, causing substantial mortality. Microplastic ingestion exhibited a dose-dependent relationship, with higher intake at high concentrations. Nematoda, turbellarians, Gastrotricha, Copepoda, Acari and Nauplii all ingested microplastics, and only Tardigrada did not. This study contributes to ongoing efforts to understand the short-term effects of microplastics at environmentally relevant concentrations in a marine meiofauna community.

Supplemental Information

Supplemental Information 1 Mean Pielou index (red line) and median (black line) of meiofauna over time

Mean Pielou index (red line) and median (black line) of meiofauna over time. Environmental control (Camb), Collection after 3 (T3), 6 (T6) and 9 (T9) days of exposure to polystyrene (PS). Different letters symbolize significant differences within each experimental time. Concentrations in particles/mL

Supplemental Information 2 Mean Shannon index (red line) and median (black line) of meiofauna over time

Mean Shannon index (red line) and median (black line) of meiofauna over time. Environmental control (Camb), Collection after 3 (T3), 6 (T6) and 9 (T9) days of exposure to polystyrene (PS). Different letters symbolize significant differences within each experimental time. Concentrations in particles/mL.

Supplemental Information 3 A: Nauplii; B: Copepode; C: Tardigrade, Internal (Ingestion-Red triangle) and external (Blue-triangle) microplastic

Internal (Ingestion-Red triangle) and external (Blue-triangle) contamination by 1 µm PS microplastic at a concentracion 107 part./mL. in marine meiofauna.

Supplemental Information 4 D: turbellarians and E: Gastrotricha, Internal (Ingestion-Red triangle) and external (Blue-triangle) microplastic

Internal (Ingestion-Red triangle) and external (Blue-triangle) contamination by 1 µm PS microplastic at a concentracion 107 part./mL. in marine meiofauna.

Supplemental Information 5 F: Acari, Internal (Ingestion-Red triangle) and external (Blue-triangle) microplastic

Internal (Ingestion-Red triangle) and external (Blue-triangle) contamination by 1 µm PS microplastic at a concentracion 107 part./mL. in marine meiofauna.

Supplemental Information 6 Nematode ingestion of microplastic over time at 107

Internal (Ingestion-Red triangle) and external (Blue triangle) contamination in Nematoda over time at a concentracion 107part./mL. Three days after exposure to microplastic (Time 3), six days after exposure to microplastic (Time 6) and nine days after exposure to microplastic (Time 9).

Supplemental Information 7 Raw data for experimental counts over time

Full data counts for meiofauna over time for all treatment of the experiment of microplastic absortion.

Supplemental Information 8 Microplastic absortion: effect on meiofauna abundance and diversity

The research leading to the results presented in this study was conducted using facilities at the Marine and Estuarine Invertebrate and Meiofauna Cultivation Laboratory (LACIMME). We express our sincere gratitude to all LACIMME members who, directly or indirectly, contributed to this research. Special thanks go to Emanuele Rodrigues Firmino, Natally Souza da Silva Costa, Letícia Pereira Pontes and Thalita Maria Santos Barbosa for their significant contributions to the continuity and advancement of this research.

Additional Information and Declarations

Competing Interests

Author Contributions

Field Study Permissions

Data Availability

The authors declare there are no competing interests.

Flávia J.L. de França conceived and designed the experiments, performed the experiments, analyzed the data, prepared figures and/or tables, authored or reviewed drafts of the article, and approved the final draft.

Tom Moens analyzed the data, authored or reviewed drafts of the article, and approved the final draft.

Renan B. da Silva performed the experiments, analyzed the data, prepared figures and/or tables, authored or reviewed drafts of the article, and approved the final draft.

Giovanna L. Pessoa performed the experiments, prepared figures and/or tables, authored or reviewed drafts of the article, and approved the final draft.

Débora A.A. França performed the experiments, prepared figures and/or tables, authored or reviewed drafts of the article, and approved the final draft.

Giovanni A.P. Dos Santos conceived and designed the experiments, analyzed the data, prepared figures and/or tables, authored or reviewed drafts of the article, and approved the final draft.

The following information was supplied relating to field study approvals (i.e., approving body and any reference numbers):

no special permission/permits were needed to collect these animals, because meiofauna are microscopic, non-pathogenic animals, field study did not involve endangered species and sampling was carried out in public beaches. Moreover, no meiofauna species are under special conservation concerns.

The following information was supplied regarding data availability:

The raw measurements are available in the Supplementary File.

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
