# Peer review of "Short-term microplastic effects on marine meiofauna abundance, diversity and community composition"

_PeerJ, doi:10.7717/peerj.17641_

## Round 0.1 · original submission · Minor Revisions

Comments included in the annotated manuscript should be addressed and it will be good if the manuscript to be sent to proofreading services.

**Language Note:** The Academic Editor has identified that the English language must be improved. PeerJ can provide language editing services - please contact us at [email protected] for pricing (be sure to provide your manuscript number and title). Alternatively, you should make your own arrangements to improve the language quality and provide details in your response letter. – PeerJ Staff

Reviewer 1 ·

Basic reporting

Short-term microplastic effects on marine meiofauna abundance, diversity and community composition

A review.

Experiments to test the effects of microplastics (MPs) on meiofauna were conducted in the laboratory. The authors prepared artificial microcosms (4 replicates for each treatment) made up of sand and meiofauna collected in nature (Brazil) to which defaunated sand enriched or not with MPs (1um polystyrene beads) was added. There were three MP concentrations: low, medium, and high. The authors indicate that the low and medium concentrations are similar to those found in nature today, while the high concentrations explore future scenarios. The short-term effects of MPs on meiofauna were evaluated by analyzing samples collected at Time 0, 3, 6, and 9 days. Several parameters, typical of the meiofauna studies, were chosen as endpoints: total density and individual taxa, values of structural parameters, etc. A count of the ingested particles was also performed.

The results indicate major effects of microplastics at low and medium concentrations compared to high concentrations. The reason for these bizarre results is unknown, but the authors try to make some speculation. Even the effects on the different parameters and between the different concentrations only sometimes appear straightforward, so much so that the authors themselves indicate that perhaps MPs have no effects on meiofauna.
Meiofaunal organisms of all major groups ingested microplastics, with the exception of Tardigrada. The authors claim the first records of PM ingestion by Gastrotricha and “Turbellaria”.

I had no difficulty reading the work but am not a native speaker. However, I have noted in the PDF that I am attaching some suggestions to improve understanding and eliminate various ambiguities.
The bibliography seems correct and exhaustive, even if I have indicated a couple of works that should be added. The figures and tables produced are essential for the article.
However, photos proving the ingestion of microplastics by Gastrotricha and “Turbellaria” seem missing. These may be included as supplementary material. BDY, “Turbellaria” is no longer a recognized taxon (since it is non-monophyletic). Therefore, the authors should indicate the specimens they believe belong to this "trash-bin" as flatworms or turbellarians (small cap).
The ambiguity of the results obtained limits the significance of the work conducted, but on the other hand, the absence of evidence to support the ingestion of microplastics in nature by meiofauna makes it challenging to support the direct danger of these pollutants on this important biotic component and transfer to higher trophic levels.
Despite the many criticizable aspects, the work can be accepted for publication if the authors consider the suggestions indicated in the attached PDF.
Sincerely

Experimental design

see pdf attched

Validity of the findings

see general comments and pdf attached

Annotated reviews are not available for download in order to protect the identity of reviewers who chose to remain anonymous.

·

Basic reporting

no comment

Experimental design

The experiment has a correct design it would be desirable to justify or argue the choice of the duration of the experiment and how they made sure not to include macrofauna organisms.

Validity of the findings

no comment

Additional comments

The article presents an interesting experiment on the effect of microplastics on meiofaunal community structure. It is well written and contributes to the knowledge of the subject.
The comments I made seek to improve the quality of the article, so I consider it important to be taken into account.

---

## Round 0.2 · accepted · Accept

I have read your track changed version and I believe you and your co-authors have addressed all comments especially from the first reviewer.

It is a pity actually for the second reviewer not to give proper or critical feedbacks that could significantly improve the manuscript. Nonetheless, the first reviewer did the job well.

I think your manuscript is ready for the next step before being published in PeerJ. The information provided by your manuscript is crucial for marine pollution studies .